# Genome destabilization-associated phenotypes arising as a consequence of therapeutic treatment are suppressed by Olaparib

**Mafuka Suzuki[1,2☺], Haruka Fujimori[1,2☺], Kakeru Wakatsuki[1], Yuya Manaka[1,3], Haruka Asai[1,3], Mai Hyodo[1,2], Yusuke Matsuno[1], Rika Kusumoto-Matsuo[1], Mitsunori Shiroishi[2], Ken-ichi Yoshioka[1]***

1 Laboratory of Genome Stability Maintenance, National Cancer Center Research Institute, Tsukiji, Chuo-ku, Tokyo, Japan, 2 Department of Biological Science and Technology, Tokyo University of Science, Niijuku, Katsushika-ku, Tokyo, Japan, 3 Department of NCC Cancer Science, Graduate School of Medical and Dental Sciences, Tokyo Medical and Dental University, Yushima, Bunkyou-ku, Tokyo, Japan

☺ These authors contributed equally to this work.
* kyoshiok@ncc.go.jp

**Data Availability Statement:** All relevant data are within the manuscript and its Supporting Information files.

## Abstract

Malignancy is often associated with therapeutic resistance and metastasis, usually arising after therapeutic treatment. These include radio- and chemo-therapies, which cause cancer cell death by inducing DNA double strand breaks (DSBs). However, it is still unclear how resistance to these DSBs is induced and whether it can be suppressed. Here, we show that DSBs induced by camptothecin (CPT) and radiation jeopardize genome stability in surviving cancer cells, ultimately leading to the development of resistance. Further, we show that cytosolic DNA, accumulating as a consequence of genomic destabilization, leads to increased cGAS/STING-pathway activation and, ultimately, increased cell migration, a precursor of metastasis. Interestingly, these genomic destabilization-associated phenotypes were suppressed by the PARP inhibitor Olaparib. Recognition of DSBs by Rad51 and genomic destabilization were largely reduced by Olaparib, while the DNA damage response and cancer cell death were effectively increased. Thus, Olaparib decreases the risk of therapeutic resistance and cell migration of cells that survive radio- and CPT-treatments.

## Introduction

DNA damage-induced cancer cell death is the mode of action for many conventional chemo-therapeutics, such as camptothecin (CPT), or radiation therapy [1–4]. This strategy is particularly effective when cancer cells are targeted preferentially compared with normal somatic cells [5, 6]. In fact, cancer cells are generally more sensitive to CPT than normal cells, because cells mutated in either ARF or p53 accumulate H2AX in response to CPT-induced damage, which results in increased γH2AX foci formation, a heightened damage response, and efficient induction of cell death [7]. Unlike cancer cells widely mutated in the ARF/p53 pathway, normal cells

**Funding:** This work was supported by AstraZeneca K.K. and Merck Sharp & Dohme Corp. (K.Y.), and partly by JSPS Kakenhi (21K12252 to K.Y. and 20K12159 to R.K.-M.). Y.M. (Yusuke Matsuno) and R.K.-M. were supported by JSPS Research Fellowships. The funders had no role in the design of the study; in the collection, analyses, or interpretation of data; in the writing of the manuscript; or in the decision to publish the results.

**Competing interests:** I have read the journal's policy and the authors of this manuscript have the following competing interests: this work was supported by AstraZeneca K.K. and Merck Sharp & Dohme Corp.This does not alter our adherence to PLOS ONE policies on sharing data and materials. The funders had no role in the design of the study; in the collection, analyses, or interpretation of data; in the writing of the manuscript; or in the decision to publish the results.

treated with CPT are quiescent because the H2AX level is usually suppressed. These differences ultimately lead to the preferential targeting of cancer cells unless resistance has been acquired. Resistant cells often become predominant following the treatment. Multiple pathways of therapeutic resistance have been identified, including the acquisition of additional mutations and epigenetic alterations that alter the cellular response to treatment [8–12], clonal selection of drug tolerant cells [13–15], and alterations to the microenvironment [10, 16]; however, the precise mechanism by which resistance is acquired to DNA damaging agents remains unclear.

Cancers generally develop as a consequence of multiple rounds of clonal evolution [17, 18], which can include a process of acquired therapeutic resistance. As recently shown, such clonal evolution can be induced by genomic destabilization triggered by replication stress-associated DSBs [19]. DSBs are the major damage associated with γ-ray irradiation [20] and CPT treatment [3, 19, 21]. Most cancer cells die in response to DSB accumulation, mainly though cell death associated with mitotic catastrophe. Some cells survive, however, expressing a senescence-associated phenotype [20]. DSBs in these cells risk genomic destabilization, which may lead to acquired resistance. While cytosolic DNA induced by genomic instability can lead to the activation of the cGAS/STING-pathway and an innate immune response [22–24], therapeutic resistance may still occur.

Genomic instability is usually induced by the introduction of replication stress-associated DSBs and caused by erroneous DNA repair [19, 25]. These DSBs are primarily repaired by homologous recombination (HR), following cell-cycle checkpoint activation by ATR [26, 27]; however, failure to induce such checkpoints, or the associated repair systems, can lead to genome instability. Therefore, it might be possible to reduce the risk of genome destabilization following cancer therapy by exogenously manipulating the DNA repair system.

In this study, we examine the effect of γ-ray irradiation and CPT treatment on genome destabilization-associated therapy resistance and cellular migration. Since PARP1 is required to mediate the recruitment of HR-factor to the site of DSBs [26–30], we further examine the effect of co-treatment with the PARP inhibitor Olaparib on the DNA damage response. Our results reveal that, while cells surviving γ-ray irradiation and CPT treatment are at increased risk of developing resistance and cellular migration, these phenotypes are largely suppressed by co-treatment with Olaparib.

## Results

### Suppression of genomic destabilization by Olaparib

CPT treatment causes replication stress-associated DSBs [3, 19, 21], which are usually targeted by HR. Genome is destabilized when those DSBs are not effectively repaired, accumulated as the intermediates of HR, and resulting in the erroneous repair [31]. Since cells surviving CPT treatment usually accumulate those DSBs, we hypothesized that suppression of HR via inhibition of PARP, a factor that mediates the induction of HR [28–30], would alter the level of genome destabilization in vitro. Initially, we examined the co-localization of γH2AX with the HR-factor Rad51 in the p53 wild-type breast cancer cell line MCF7 in the presence and the absence of Olaparib (Fig 1A). As expected, γH2AX colocalized with Rad51 in discrete foci in the absence of Olaparib; however, the number of these foci was significantly reduced following Olaparib treatment. Next, we examined γH2AX foci for the presence of 53BP1, a protein that accumulates at DSB sites. The majority of γH2AX foci contained 53BP1 both in the presence and absence of Olaparib (85% and 75%, respectively) (Fig 1B), suggesting that these foci were associated with DSBs. Together, these results show that replication stress-associated DSBs are recognized by Rad51 in a PARP-dependent manner. Replication stress-associated DSBs are often carried over into M-phase, inducing genome instability [32]. To monitor this in our

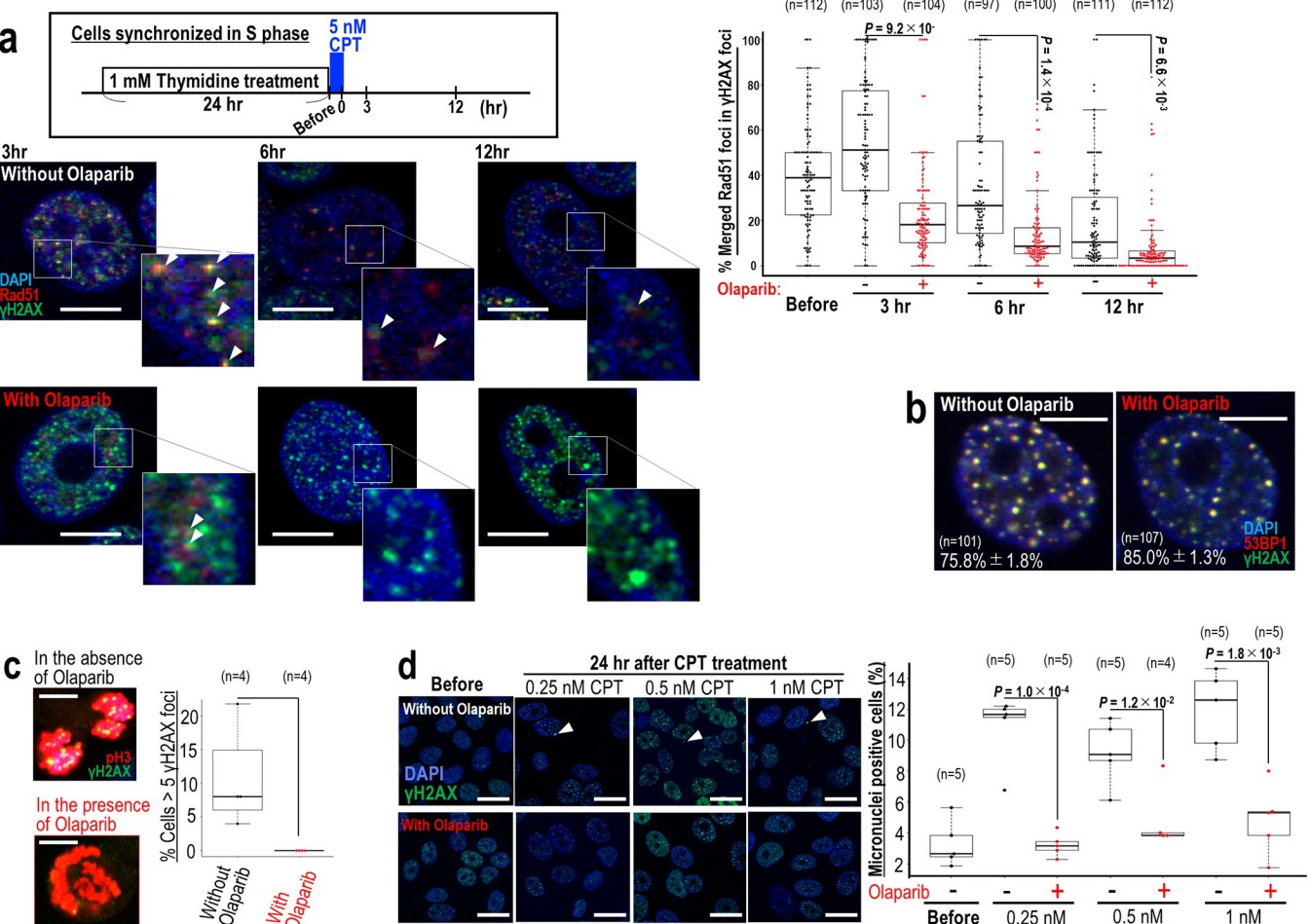

**Fig 1. DSB recognition is altered, leading to suppression of genome destabilization-associated phenotypes.** (**a**) MCF7 cells synchronized in S phase were pulse treated by CPT for 1 hr and further cultivated for 12 hr, as shown in the upper panel. Foci containing γH2AX and Rad51 were detected by immunofluorescence. Representative images are shown in the left panel and quantification is shown in the right panel. (**b**) MCF7 cells treated as in (**a**) were stained with anti-γH2AX and -53BP1 antibodies after 12 hr. (**c**) MCF7 cells treated as in (**a**) were further cultivated for 12 hr and stained for γH2AX foci and p-H3. (**d**) MCF7 cells were treated for 24 hr in the presence of 0.25, 0.5, and 1 nM CPT, and micronuclei were identified by immunofluorescence staining for γH2AX and DAPI (n numbers are indicated on graph). Scale bars in immunofluorescence images, 10 μm. Bars in box and whisker plots show mean ± s.d. Two-tailed Welch's t-test was used for statistical analysis.

system, we examined the number of γH2AX foci carried over into M-phase, visualized by immuno-staining for H3 phosphorylated at S10, in Olaparib-treated and non-treated cells. Many M-phase γH2AX foci were observed in CPT-treated cells and these foci were suppressed in the presence of Olaparib (Fig 1C). Consistent with these observations, the number of micronuclei associated with genomic instability was elevated following CPT treatment in the absence of Olaparib compared with Olaparib-treated cells (Fig 1D). These data suggest that replication stress-associated DSBs are recognized in a PARP-dependent manner by Rad51, and this recognition is associated with genomic destabilization.

## CPT-induced cellular migration is suppressed by Olaparib

It has previously been shown that micronuclei activate the cGAS/STING innate immune response pathway, resulting in elevated cell migration [33]. We, therefore, examined the effect of CPT on the formation of cGAS-positive micronuclei and cell migration in the presence of

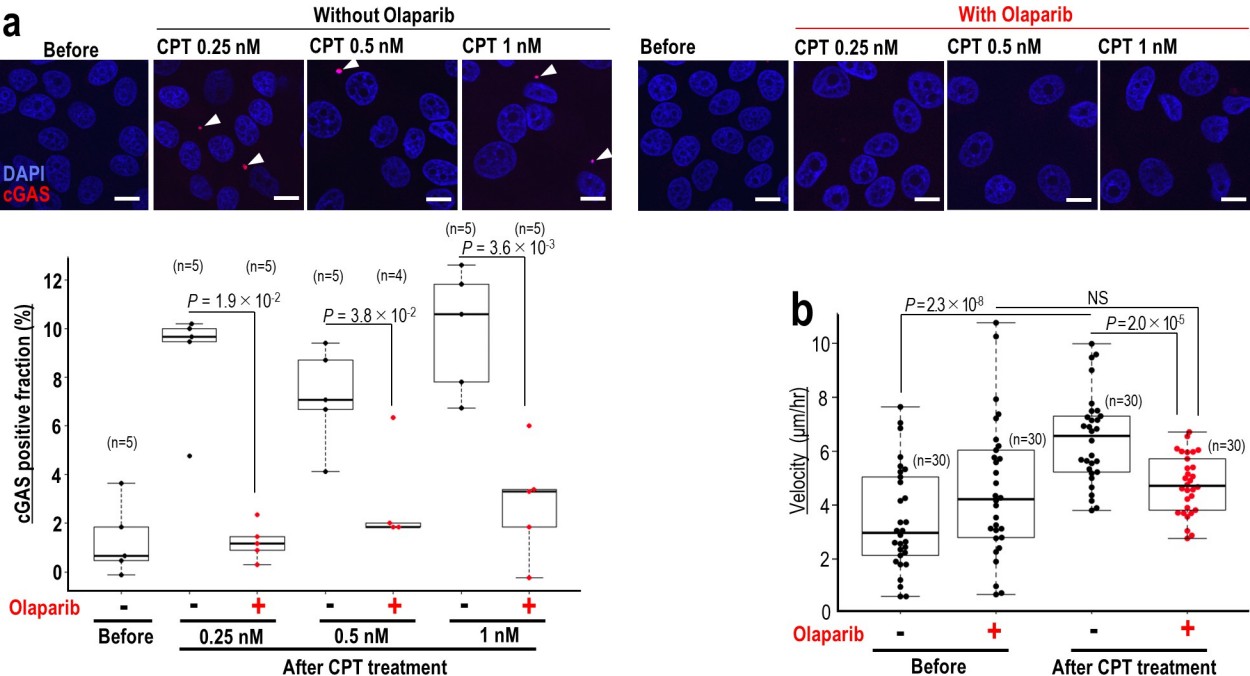

**Fig 2. cGAS activation is suppressed by Olaparib, resulting in the suppression of cellular migration.** (**a**) MCF7 cells treated with CPT were cultivated for 24 hr, and cGAS-positive cells were identified by immunofluorescence. (**b**) MCF7 cells were treated as in (**a**) with 0.5 nM CPT and monitored by time lapse imaging (n numbers are indicated on graph). Migration rates are shown as the velocity 24 to 48 hr post CPT treatment. Scale bars in images, 10 μm. Bars show means ± s.d. Two-tailed Welch's t-test was used for statistical analysis.

Olaparib. CPT treatment largely increased the number of cells with cGAS-positive micronuclei in the absence of Olaparib, while co-treatment with Olaparib significantly reduced the number of these micronuclei to a level similar to non-CPT-treated control (Fig 2A). Consistent with these observations, cGAS/STING-associated cellular migration was elevated following CPT treatment; a phenotype suppressed in the presence of Olaparib (Fig 2B). These results indicate that Olaparib suppresses genome destabilization, cGAS/STING activation, and associated cellular migration.

## Radiation-associated genome destabilization phenotypes are suppressed by Olaparib

Radiation therapy is used in the treatment of many cancers; however, radioresistance can develop. A recent study revealed that although multiple types of DNA damage are induced by irradiation, persistent DSBs are induced in association with replication stress, and their erroneous repair further increases the risk of genome stability [20]. We next examined the effect of γ-ray-induced DSBs on cell survival and micronuclei formation in the presence of Olaparib (Fig 3). Micronuclei induced by 10 Gy irradiation were significantly suppressed in the presence of Olaparib (Fig 3A), while γ-ray-induced γH2AX/53BP1 foci were unaffected (Fig 3B). To investigate the effect of γ-ray irradiation on the cGAS/STING-pathway activation, we examined the number of cGAS-positive micronuclei (Fig 3C). As expected, γ-ray irradiation-induced cGAS-positive micronuclei were significantly reduced in the presence of Olaparib. Similar effects were observed in HeLa and SW480 cells (S1 Fig in S1 File). Consistent with this, cGAS/STING-associated phenotypes, such as the induction of Ifnβ, Ifit1, and Ifit3 expression and cellular migration, elicited by γ-ray irradiation were significantly suppressed by Olaparib

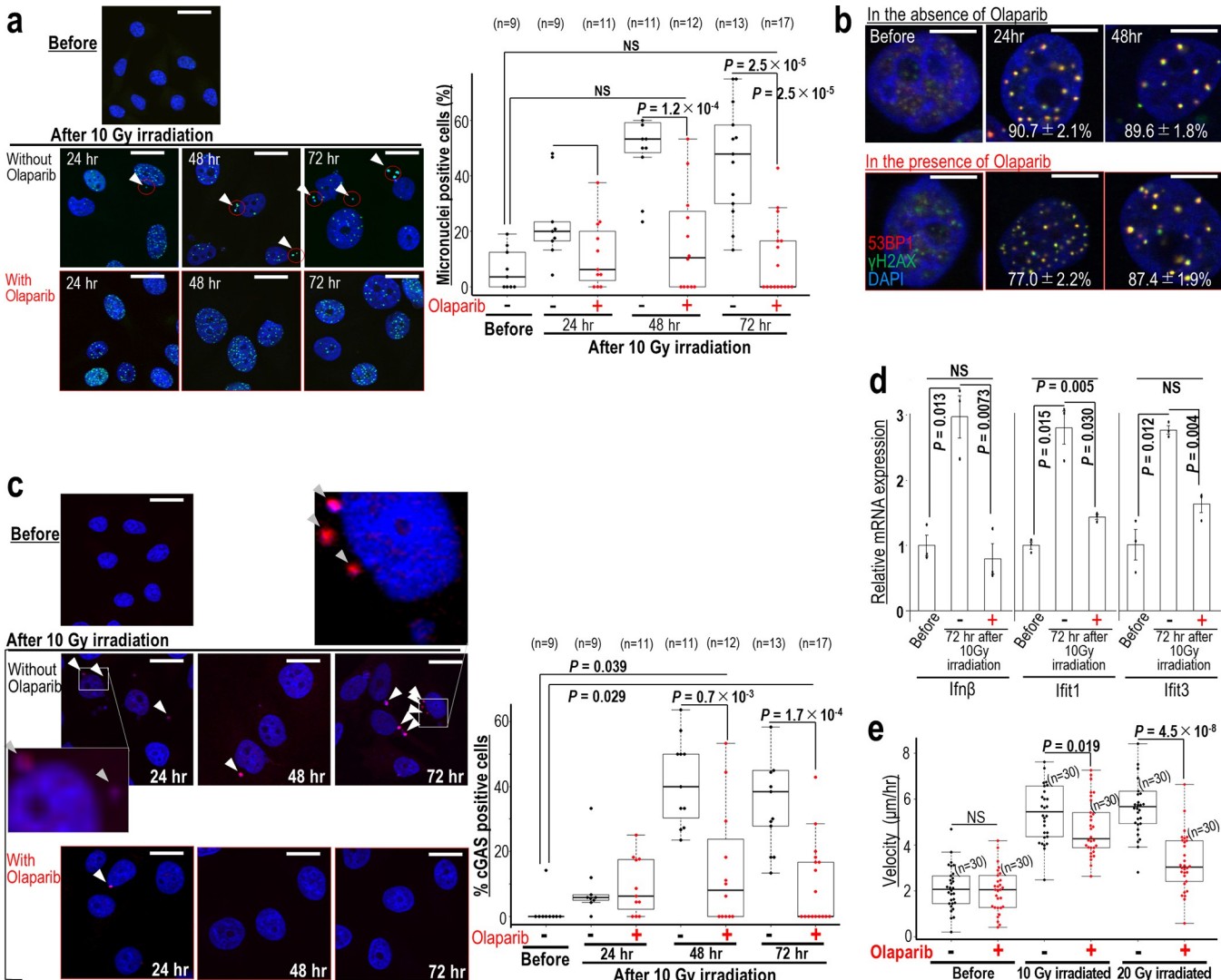

**Fig 3. Genomic destabilization-associated phenotypes induced by radiation are suppressed by Olaparib.** (**a**, **b**) MCF7 cells were irradiated with γ-ray (10 Gy) in the presence or absence of Olaparib. Micronuclei were identified by immunofluorescence staining with a γH2AX antibody and DAPI (**a**). Arrowheads show micronuclei. Scale bars in images, 30 μm. Bars show means ± s.d. Two-tailed Welch's t-test was used for statistical analysis. Foci containing γH2AX and 53BP1 were detected by immunofluorescence (**b**). Scale bars in images, 10 μm. (**c**) MCF7 cells were examined 24–72 hr after 10 Gy γ-ray irradiation in the presence or absence of Olaparib. Micronuclei were identified by immunofluorescence staining with a γH2AX antibody and DAPI. Arrowheads show cGAS-positive micronuclei. Scale bars in images, 30 μm. Bars show means ± s.d. Two-tailed Welch's t-test was used for statistical analysis. (**d**) MCF7 cells were treated as in (**c**) and the expression of *Ifnβ*, *Ifit1*, and *Ifit3* was determined (n = 3 independent experiments). (**e**) MCF7 cells were treated as in (**c**) and monitored by time lapse imaging (n numbers are indicated on the graph). Migration rates are shown as the velocity 24 to 48 hr post γ-ray irradiation. Two-tailed Welch's t-test was used for statistical analysis.

(Fig 3D and 3E). Together, these data suggest that DSBs, induced by either CPT or γ-ray irradiation, increase the risk of genomic destabilization, resulting in micronuclei formation, cGAS/STING-pathway activation, and cell migration. Importantly, those genomic destabilization-associated effects are significantly suppressed in the presence of Olaparib.

## Chemo- and radioresistance is suppressed by Olaparib

Next, we wanted to examine the effect of Olaparib on acquired resistance following chemo- and radio-treatments. MCF7 cells were treated with CPT in the presence and absence of

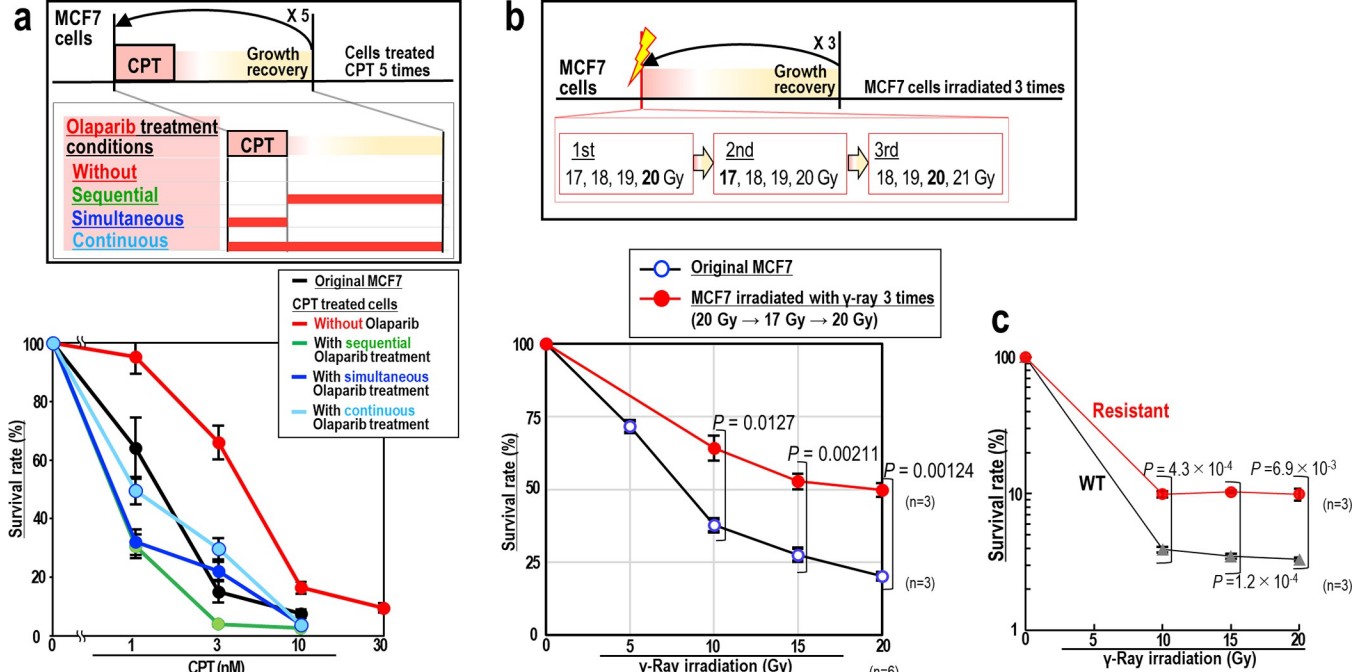

**Fig 4. Radio- and CPT-resistance is suppressed by Olaparib. (a)** MCF7 cells were treated with CPT, as shown in the upper panel, to induce resistance. Untreated and treated cells were subsequently treated with increasing concentrations of CPT, and their survival was assessed (n = 3 independent experiments). Graphs show mean survival rates ± s.d. from three independent experiments. **(b,c)** MCF7 cells were treated with γ-ray irradiation, as shown in the upper panel, to induce resistance (n = 3 independent experiments). Untreated and treated cells were subsequently irradiated by γ-ray, and their survival was assessed by cell counting (**b**) or MTS assay (**c**). Graphs show mean survival rates ± s.d. from three independent experiments. Two-tailed Welch's t-test was used for statistical analysis.

Olaparib, following three different treatment schedules: sequential, simultaneous, and continuous treatment (Fig 4A). Cells were treated with CPT for 5 days under conditions that allowed 0.5–2% of the cells to survive. Surviving cells were continuously maintained until their growth recovered, after which the cells were subjected to the next cycle of CPT treatment. After 5 cycles of CPT treatment, the resistance of the cells to CPT was compared. Co-treatment with Olaparib suppressed acquired resistance to CPT irrespective of how Olaparib was administered (Fig 4A), consistent with the finding that genomic destabilization induced by CPT is suppressed by Olaparib. Intriguingly, administration of Olaparib sequentially following CPT suppressed resistance to a level similar to that of the original MCF7 control, implying that this suppression by Olaparib is effective even in the presence of pre-existing replication stress-associated DSBs.

Next, we examined the effect of Olaparib on acquired resistance to γ-ray irradiation (Fig 4B and 4C). MCF7 cells were irradiated with multiple doses of γ-ray irradiation (17–21 Gy) in the presence or absence of Olaparib, after which 0.5–5% of the cells were still alive at 1 week after irradiation. The surviving cells were continuously maintained until growth recovered and were then subjected to the next cycle of γ-ray irradiation. As expected, cells irradiated three times in the absence of Olaparib showed increased resistance to radiation (Fig 4B and 4C). By contrast, the cells that survived irradiation in the presence of Olaparib showed a flattened and enlarged morphology without recovery of cell growth; therefore, we failed to obtain cells that had been irradiated three times in the presence of Olaparib. These results indicate that any induced resistance to γ-ray irradiation is suppressed in the presence of Olaparib.

### The DNA damage response is enhanced following radiation- and CPT treatment

To examine the effect of Olaparib on the response to DNA damage induced by γ-ray irradiation, we analyzed γH2AX signal 24 hr post irradiation (10 and 20 Gy) (Fig 5A). Since the level of damage in some cells was too severe to count foci number, especially in cells irradiated with 20 Gy in the presence of Olaparib, γH2AX signal intensities and the apparent number of foci were measured in individual nuclei (Fig 5A). Both γH2AX foci number and signal intensities were significantly increased when Olaparib was administered, with a 3-fold increase in γH2AX intensity seen 24 hr after 20 Gy irradiation in the presence of Olaparib, compared with cells treated with radiation alone. In the presence of Olaparib, γH2AX staining appeared to be pan-nuclear, which is usually a marker for cells undergoing apoptosis. Together, these results indicated that the DNA damage response was enhanced by Olaparib. An identical damage response enhancement by Olaparib was found in p53-deficient cancer cell lines HeLa and SW480 treated with Olaparib (S1 Fig in S1 File). CPT-treatment elicited a similar response (Fig 5B): both γH2AX intensity and foci numbers were increased 2-fold at 12 hr post CPT treatment in the presence of Olaparib compared with CPT treatment alone. We conclude that Olaparib administration significantly enhances γH2AX foci formation in response to replication stress-associated DSBs induced by both γ-ray irradiation and CPT treatments.

We further tested the effect of Olaparib on the induction of cancer cell death following radio- and chemo-treatments (Fig 5C and 5D). Olaparib significantly increased cell death following γ-ray irradiation in both direct counting and colony formation assays (Fig 5C). Similar results were seen following CPT treatment (Fig 5D). Thus, cancer cell death caused by γ-ray irradiation and CPT treatment can be efficiently increased by Olaparib. Together, these data suggest that Olaparib treatment during radio- or CPT-treatment suppresses genomic destabilization, cell migration, and therapy resistance, in association with a heightened DNA damage response and increased cancer cell death.

## Discussion

Radiotherapy and many chemotherapies induce cancer cell death by inducing DNA damage. In the current study, we have revealed that cells surviving this damage are at an increased risk of genomic destabilization, ultimately leading to cellular migration and clonal evolution of cells with acquired resistance. The risk of cellular migration has been shown to be elevated through the activation of the cGAS/STING pathway in response to cytosolic DNA produced as a consequence of genomic destabilization. These effects were observed following γ-ray irradiation and CPT treatment, suggesting that cancers treated with radiotherapy and/or CPT chemotherapy are at an increased risk of cell migration and the acquisition of resistance. Importantly, such genomic destabilization-associated risks can be effectively suppressed by the presence of Olaparib through the modulation of DNA repair pathways, damage response enhancement, and enhanced induction of cancer cell death.

Since therapeutic resistance often occurs after therapy [22, 34, 35], it is important to develop strategies that suppress this resistance. In the current study, simultaneous treatment with an anti-cancer therapy and Olaparib reduced the risk of this resistance developing. However, the overall clinical benefit still requires careful consideration, because the cGAS/STING pathway, which is induced in response to cytosolic DNA generated by genomic instability, is also associated with immune responses that have anti-tumor effects [23, 36]. For example, in BRCA1-mutated cancer cells, the killing effect of Olaparib treatment is observed in association with cGAS/STING pathway activation [37]. Thus, to determine the overall clinical benefit, further study into the innate immune response and acquired resistance and how these are balanced in each cellular context is required.

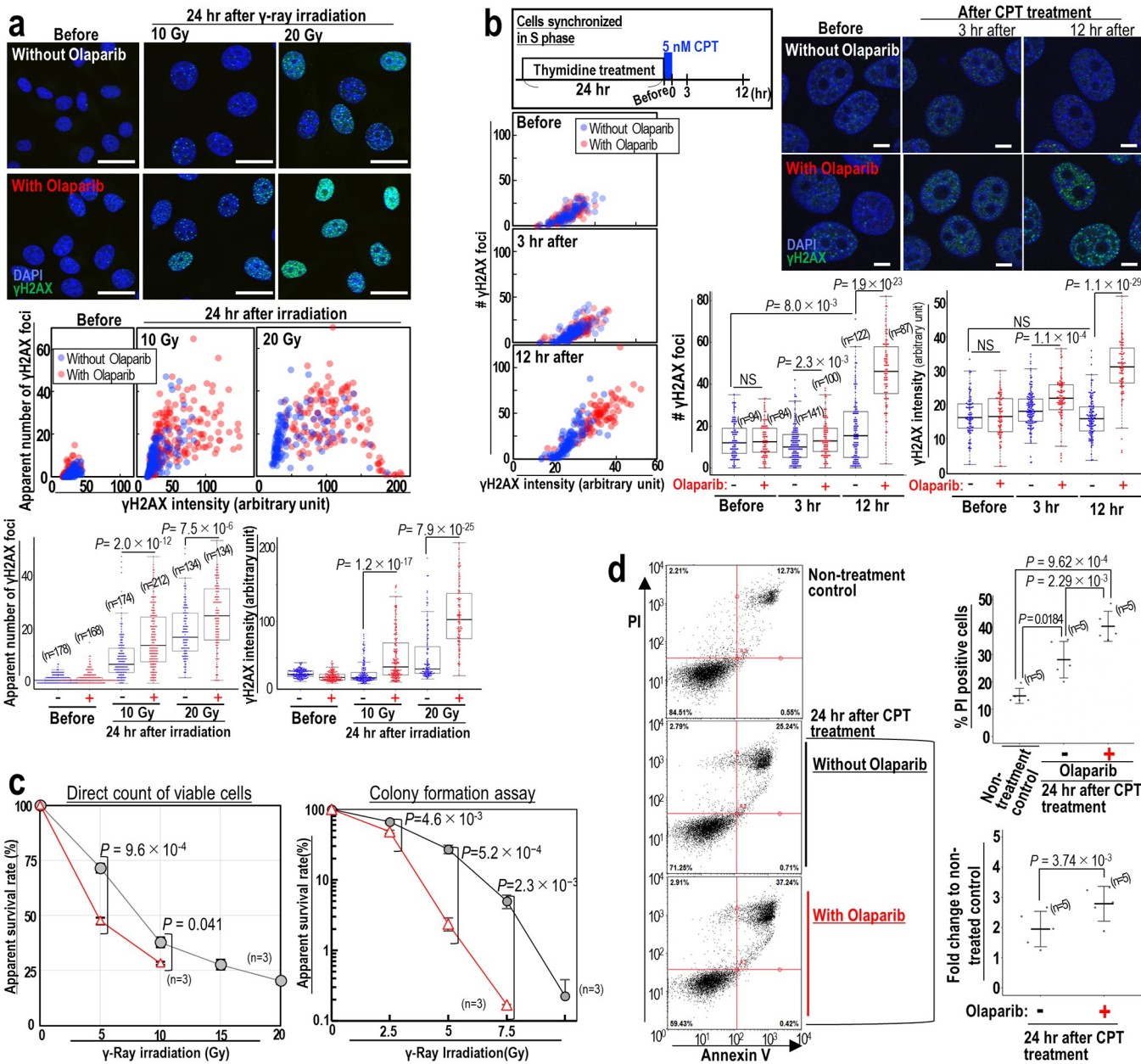

**Fig 5. Sensitized DSB responses in the presence of Olaparib.** (**a**) MCF7 cells were irradiated with 10 and 20 Gy γ-ray, and cultivated for 24 hr. γH2AX intensity and number of γH2AX foci are plotted together and separately in the absence (blue dots) and the presence (red dots) of Olaparib. Scale bars in images, 30 μm. Bars show means ± s.d. Two-tailed Welch's t-test was used for statistical analysis. (**b**) MCF7 cells synchronized in S phase were pulse treated with CPT for 1 hr and further cultivated for 12 hr, as described in the upper panel. γH2AX intensity and number of γH2AX foci are plotted together and separately in the absence (blue dots) and presence (red dots) of Olaparib. Scale bars in images, 10 μm. Bars show means ± s.d. Two-tailed Welch's t-test was used for statistical analysis. (**c**) Cell survival calculated by direct counting of surviving cells 5 days post γ-ray irradiation in the presence or absence of Olaparib (red triangle and black circles, respectively). The numbers of surviving cells irradiated with 15 and 20 Gy in the presence of Olaparib were too low and therefore could not be counted. Cell survival was similarly estimated by colony formation assay. Bars show means ± s.d. Two-tailed Welch's t-test was used for statistical analysis. (**d**) MCF7 cells treated as in (**b**) in the presence and absence of Olaparib. Cell death was analyzed by double staining of PI and annexin V (n numbers are indicated on graph). Bars show means ± s.d. Two-tailed Welch's t-test was used for statistical analysis.

## Genomic destabilization-associated risks elevated with therapeutic treatment

Cancer development steps usually progress through multiple rounds of clonal evolution. Genomic destabilization can lead to such clonal evolution [19]. In the current study, we show that such genomic destabilization is also associated with increased cellular migration, which is further associated with metastasis, and increased resistance acquisition. Given that malignancy is widely associated with therapeutic resistance and metastasis, it is possible that anti-cancer therapies that induce replication stress-associated DSBs are simultaneously associated with increased risk of malignancy, highlighting the importance of suppression of genomic destabilization.

## Neutralization of genomic destabilization-associated risks

A fraction of cancer cells survive the DNA damage caused by γ-ray irradiation and CPT treatment, and genomic destabilization-associated phenotypes are likely to appear as a consequence of these treatments. It is critical to understand how clonal evolution, cellular migration, and other genomic destabilization-associated phenotypes can be suppressed. Genomic destabilization is generally triggered by erroneous repair of replication stress-associated DSBs, especially when HR is ineffective [19, 20]. HR induction is primarily mediated by PARP1 [28–30], and DSB recognition by HR factors is largely suppressed by Olaparib. Indeed, our results indicated that simultaneous treatment with CPT (or γ-ray) and Olaparib suppresses Rad51 recognition of DSBs, resulting in the suppression of genomic destabilization, resistance acquisition, and cell migration. The current study also revealed that simultaneous treatment with Olaparib is also associated with a stronger DNA damage response, resulting in increased cancer cell death. This is consistent with a previous study showing the radio-sensitization by PARP inhibitor [38]. Such sensitization might be induced in association with the changes that occur in the resulting damage response; while ATR Ser/Thr protein kinase is activated in response to replication stress-associated DSBs that are repaired by HR, ATM kinase is activated in response to DSBs that are not repaired by HR. Since histone H2AX is transiently induced upon ATM activation, this results in a heightened damage response and increases in γH2AX foci formation, which in turn result in efficient induction of cancer-cell death [39].

Our results revealed that the genomic instability associated with cancer therapies is a risk factor for the development of resistance against these therapies and for cell migration. Importantly, we show that this risk can be suppressed by manipulation of DNA repair pathways associated with genomic instability. The risk of resistance after CPT treatment and/or γ-ray irradiation can be suppressed by Olaparib treatment.

## Methods

### Cell culture, cell viability assays, and gene expression assay

MCF7, HeLa, and SW480 cells were obtained from ATCC. Immortalized MEFs (mouse embryonic fibroblasts) used in the current study were prepared as described in a previous study [19], in which primarily MEFs were originally prepared from mouse embryos [40]. MEFs used in this experiment were prepared in a specific pathogen-free environment at the National Cancer Center Research Institute (Tokyo, Japan) animal facility according to the institutional guidelines, and with the approval of the Japan National Cancer Center Animal Ethics Committee (B104M2-18, A364M2-18). Cervical dislocation was used for euthanasia. Cells were cultivated in Dulbecco's Modified Eagle's Medium (Nakarai) supplemented with 10% (v/v) fetal calf serum (Gibco).

Cell viability assays were performed by direct counting of surviving cells, colony formation assay, and MTS assay. Direct count of surviving cells was performed after visualization with Acridine Orange (Wako) on a laser microscope (Keyence BZ-X800). Colony formation assays were performed after γ-ray irradiation by counting the resulting colonies, in which 500–2,000 cells were seeded. MTS assay was performed using the CellTiter 96 AQueous one solution reagent (Promega) and GloMax (Promega). While multiple assays were used to examine cell viability, direct counting of surviving cells was used in the main, in part because MTS assay does not allow for direct detection of viable cells and the efficiency of colony formation assays was affected by cell migration.

For the analysis of gene expression, RNA was extracted from cells using ReliaPrep RNA Miniprep Systems (PROMEGA) and was subjected to reverse transcriptase using PrimeScript™ RT Master Mix (Takara Bio). The resulting cDNA was used as a template for real time PCR using iTaq universal SYBR Green Supermix (Bio-Rad) with the CFX96 Touch Deep Well Real-Time PCR Detection System (Bio-Rad). PCR Primers for IFNβ, IFT1, and IFIT3 are listed in S1 Table in S1 File, with β-actin used as a normalization control.

## Cell imaging following DNA damage

DNA damage was induced by treating cells with CPT (Sigma) or by $^{137}$Cs irradiation in a Gammacell 40 Exactor (Best Theratronics). Thymidine (Sigma) and Olaparib (AstraZeneca) were used at concentrations of 0.5 mM and 1 μM, respectively. For immunofluorescence studies, cells were fixed in 4% paraformaldehyde, permeabilized with 0.1% Triton X-100/PBS, and blocked in 2% goat serum in PBS containing 0.3% Triton X-100. Immunofluorescence was conducted, using the primary and secondary antibodies indicated below, on a confocal laser microscope (Olympus FV10i and Leica SP8).

To examine micronuclei formation following DNA damage, cells were treated with CPT and/or irradiated with γ-ray at the indicated doses. Micronuclei were visualized, following DAPI and γH2AX staining, and counted at the indicated time.

To examine cell migration, time lapse imaging was performed on an IncuCyte zoom system (Sartorius). Images were acquired once per hour, and cellular migration was assessed using a manual tracking tool in ImageJ. Average migration velocity following irradiation was measured 24–48 hr post treatment, while the velocity of unirradiated control cells was measured for 10 hr.

Statistical significance was calculated by two-tailed Welch's t-tests and P-values are shown on dot plots and bar graphs.

## Antibodies

The following antibodies were used in this study: γH2AX (JBW301, Upstate Biotechnology; 9718, Cell Signaling Technology), p-H3 (H3 phosphorylated at Ser 33) (NB100-544, Noxus Biologicals), 53BP1 (PC712, Merck), Rad51 (ab133534, Abcam), cGAS (D1D3G, Cell Signaling Technology), mouse IgG-Alexa Fluor 488 (A-11001, Thermo Fisher), and rabbit IgG-Alexa Fluor 594 (A-11012, Thermo Fisher).

## Supporting information

**S1 File. Contains all the supporting tables and figures.**
(PDF)

## Acknowledgments

The authors thank Dr. H. Aoyama for critical discussion of the manuscript.

## Author Contributions

**Conceptualization:** Haruka Fujimori, Ken-ichi Yoshioka.

**Data curation:** Mafuka Suzuki, Haruka Fujimori, Kakeru Wakatsuki, Yuya Manaka, Haruka Asai, Mai Hyodo, Yusuke Matsuno, Rika Kusumoto-Matsuo, Ken-ichi Yoshioka.

**Formal analysis:** Mafuka Suzuki, Haruka Fujimori, Kakeru Wakatsuki, Yuya Manaka, Haruka Asai, Mai Hyodo, Yusuke Matsuno, Rika Kusumoto-Matsuo.

**Funding acquisition:** Ken-ichi Yoshioka.

**Investigation:** Mafuka Suzuki, Haruka Fujimori, Kakeru Wakatsuki, Yuya Manaka, Haruka Asai, Mai Hyodo, Rika Kusumoto-Matsuo.

**Methodology:** Mafuka Suzuki, Haruka Fujimori, Kakeru Wakatsuki, Ken-ichi Yoshioka.

**Project administration:** Ken-ichi Yoshioka.

**Resources:** Ken-ichi Yoshioka.

**Supervision:** Mitsunori Shiroishi, Ken-ichi Yoshioka.

**Visualization:** Mafuka Suzuki, Haruka Fujimori, Kakeru Wakatsuki, Yuya Manaka, Haruka Asai, Mai Hyodo, Yusuke Matsuno, Rika Kusumoto-Matsuo, Ken-ichi Yoshioka.

**Writing – original draft:** Ken-ichi Yoshioka.

**Writing – review & editing:** Ken-ichi Yoshioka.

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
