## [Decision Letter · Decision Letter 0]

28 Nov 2022

PONE-D-22-25618Genome Destabilization-Associated Phenotypes Arising as a Consequence of Therapeutic Treatment are Suppressed by OlaparibPLOS ONE

Dear Dr. Yoshioka,

Thank you for submitting your manuscript to PLOS ONE. After careful consideration, we feel that it has merit but does not fully meet PLOS ONE’s publication criteria as it currently stands. Therefore, we invite you to submit a revised version of the manuscript that addresses the points raised during the review process.

We look forward to receiving your revised manuscript.

Kind regards,

Alvaro Galli

Academic Editor

PLOS ONE

Journal Requirements:

“This work was supported by AstraZeneca K.K. and Merck Sharp & Dohme Corp. (K.Y.), and partly by JSPS Kakenhi (21K12252 to K.Y. and 20K12159 to R.K.-M.). Y.M. (Yusuke Matsuno) and R.K.-M. were supported by JSPS Research Fellowships.”

“This work was supported by AstraZeneca K.K. and Merck Sharp & Dohme Corp. (K.Y.), and partly by JSPS Kakenhi (21K12252 to K.Y. and 20K12159 to R.K.-M.). Y.M. (Yusuke Matsuno) and R.K.-M. were supported by JSPS Research Fellowships. The funders had no role in the design of the study; in the collection, analyses, or interpretation of data; in the writing of the manuscript; or in the decision to publish the results.”

3.  Thank you for stating the following in the Competing Interests/Financial Disclosure * (delete as necessary) section:

“This work was supported by AstraZeneca K.K. and Merck Sharp & Dohme Corp. (K.Y.), and partly by JSPS Kakenhi (21K12252 to K.Y. and 20K12159 to R.K.-M.). Y.M. (Yusuke Matsuno) and R.K.-M. were supported by JSPS Research Fellowships. The funders had no role in the design of the study; in the collection, analyses, or interpretation of data; in the writing of the manuscript; or in the decision to publish the results.”

We note that you received funding from a commercial source: [Name of Company]

“I have read the journal's policy and the authors of this manuscript have the following competing interests: this work was supported by AstraZeneca K.K. and Merck Sharp & Dohme Corp. The funders had no role in the design of the study; in the collection, analyses, or interpretation of data; in the writing of the manuscript; or in the decision to publish the results.”

Reviewers' comments:

Reviewer's Responses to Questions

**Comments to the Author**

1. Is the manuscript technically sound, and do the data support the conclusions?

Reviewer #1: Yes

Reviewer #2: Yes

2. Has the statistical analysis been performed appropriately and rigorously? 

Reviewer #1: Yes

Reviewer #2: Yes

3. Have the authors made all data underlying the findings in their manuscript fully available?

Reviewer #1: Yes

Reviewer #2: Yes

4. Is the manuscript presented in an intelligible fashion and written in standard English?

Reviewer #1: Yes

Reviewer #2: Yes

5. Review Comments to the Author

Reviewer #1: In this manuscript the authors examine how the PARP inhibitor Olaparib suppresses cancer cells that developed resistance to camptothecin and radiation-induced double stranded breaks. While the data supports the observation that Olaparib activates the cGAS pathway, the evidence was only limited to the cytosolic cGAS observed in immunofluorescence microscopy. The conclusions about combining Olaparib and campthothecin for effective cancer cells killing are not unexpected since other groups have previously published that Olaparib disrupts ovarian and colorectal cancer cell lines (Ding et al Cell Rep, 2018; 25(11): 2972–2980.e5, Miura et al Radiat Oncol, 2012; 7, 62). The observations that Olaparib disrupts gamma H2AX and 53BP1 foci co-localisation suggest that PARP inhibition activates cGAS STING pathway that contributes to tumor cell death. It has recently been shown that the Fanconi Anemia DNA repair pathway bridges the HR and NHEJ pathway and would be useful to investigate further. Overall, I think the results will be useful to the field and may stimulate additional experiments by other investigators. However my enthusiasm is tempered since the major conclusions were already previewed in other published papers. If the authors could explain which DNA damage pathways (BRCA1 or 53BP1 mediated) that could be more exciting. Otherwise the observations may be more appropriate for another journal.

Major comments:

1. Olaparib was previously shown to disrupt cGAS STING pathway and in BRCA1 deficient cells (Miura et al Radiat Oncol, 2012; Ding et al Cell Rep, 2018). The Miura et al, Radiat Oncol paper should be referenced in this manuscript.

2. PARP inhibitors are a classical example of a synthetic lethality relationship of diverging DNA repair mechanisms, and it is possible that Olaparib kills cancer cells after other HR pathways were compromised (eg. cells with stalled replication forks due to defective BRCA1 pathway). It was unclear which homologous repair pathway (eg BRCA1/Fanconi Anemia pathway) or non homologous end joining pathway (53BP1) that Olaparib affects. Cell imaging following Olaparib treatment with gH2AX and Rad51 staining should be repeated in 53BP1 and/or BRCA1/FANCD2 knock out cell lines. Alternatively, the authors could repeat the immunofluorescence imaging with BRCA1/FANCD2 to determine which pathway Olaparib affects.

3.Fig 5a – higher resolution of the figure containing the graph is needed. The number P=.. were ineligible. Fig 5c was missing two points for the gamma-ray irradiation (15 and 20 Gy).

4. Page 17 line 305 – there is already a paper that describes Olaparib association with BRCA or Fanconi Anemia pathway, please cite Ding et al Cell Rep. 2018 Dec 11; 25(11): 2972–2980.e5 as Olaparib has been shown to trigger cGas in BRCA1-deficient cells, which suggests it efficacy in killing cancer cells.

5. Please add the concentration of Olaparib as it was not described anywhere in the manuscript or methods.

Minor comments:

1. page 18 line 308 is missing a reference after "..presumably by non-homologous end joining".

2. page 18 line 313 please add a sentence or two on how this work will bring impact to the cancer therapeutics or DNA damage fields.

3. Figure 4b remove the "With or without Olaparib" figure legend as the data was not shown in the figure. Alternatively, the authors could clarify that the graph on Fig 4b was without Olaparib.

4. Figures 2a and 3c was unclear. Higher resolution images are required to see the foci.

Reviewer #2: Suzuki et al. presented a concise study on the suppression of radio- and CPT surviving cells by Olaparib co-treatment. Although the enhanced anti-tumour activity of Olaparib and CPT or gamma-rays was presented several years ago (Miura et al 2012 doi: 10.1186/1748-717X-7-62), the authors are addressing important aspect of simultaneous treatment that enhance elimination of resistant cancer cells, using MCF7 and to some extent HeLa and SW480 models. Overall, the text is well-written and data presented on figures are complete. Thus, I recommend the manuscript for publication, however, for increasing accessibility of the manuscript content I suggest some minor modifications of text, listed below.

1) The quality of figures (at least in my version, even after download) was low and should be enhanced in final version – the authors should clearly export the files at least in 300 DPI.

2) Fig 4 b and c – the authors claim that ‘no cells irradiated three times were obtained in the presence of Olaparib, because growth failed to recover in the presence of Olaparib’. This description is unclear to me and should be extended / commented. It is a bit disturbing that absolutely no cells were found in the presence of Olaparib. I convince the authors to rewrite the description of this results to explain the observed strong effects.

3) Lines 272-273 require edition as it reads strange in current version.

4) Lines 302-304 The authors should state here that it is simultaneous treatment of CPT or gamma-rays with Olaparib.

5) The final sentence of Discussion (lines 311-313) should be followed by a major conclusion on the findings described in this work to leave the reader with a clear message.

6) Methods – line 319 – used or prepared?

6. PLOS authors have the option to publish the peer review history of their article (what does this mean?). If published, this will include your full peer review and any attached files.

Reviewer #1: **Yes: **Winnie Tan

Reviewer #2: No

---

## [Author Response · Author response to Decision Letter 0]

10 Jan 2023

Reviewer #1: In this manuscript the authors examine how the PARP inhibitor Olaparib suppresses cancer cells that developed resistance to camptothecin and radiation-induced double stranded breaks. While the data supports the observation that Olaparib activates the cGAS pathway, the evidence was only limited to the cytosolic cGAS observed in immunofluorescence microscopy. The conclusions about combining Olaparib and campthothecin for effective cancer cells killing are not unexpected since other groups have previously published that Olaparib disrupts ovarian and colorectal cancer cell lines (Ding et al Cell Rep, 2018; 25(11): 2972–2980.e5, Miura et al Radiat Oncol, 2012; 7, 62). The observations that Olaparib disrupts gamma H2AX and 53BP1 foci co-localisation suggest that PARP inhibition activates cGAS STING pathway that contributes to tumor cell death. It has recently been shown that the Fanconi Anemia DNA repair pathway bridges the HR and NHEJ pathway and would be useful to investigate further. Overall, I think the results will be useful to the field and may stimulate additional experiments by other investigators. However my enthusiasm is tempered since the major conclusions were already previewed in other published papers. If the authors could explain which DNA damage pathways (BRCA1 or 53BP1 mediated) that could be more exciting. Otherwise the observations may be more appropriate for another journal.

Response: We thank the reviewer for these valuable comments and have revised the manuscript according to the reviewer’s suggestions. We have addressed the issues raised by the reviewer by adding references to the suggested studies. Although those studies are certainly important, they differ from our study is several respects. We would like to explain the differences as follows.

First, Ding et al. studied STING pathway activation in BRCA1-mutated tumor cells that are sensitive to the PARP inhibitor Olaparib (Ding et al. Cell Rep, 2018; 25(11): 2972–2980.e5). By contrast, we studied the effects of Olaparib on BRCA1-wt cells that are not sensitive to Olaparib. Since cells treated with CPT and/or irradiated with gamma-rays are often subject to genomic instability that increases the risks of cell migration and treatment resistance, we believed it was important to suppress these risks. Our results revealed that the increase risk of resistance associated with genomic instability is suppressed by Olaparib, which to the best of our knowledge has not been reported in previous studies.

In addition, Miura et al. studied the effects of Olaparib as a sensitizer of CPT and gamma-ray irradiation by monitoring cell survival rates and gammaH2AX/Rad51 foci (Miura et al Radiat Oncol, 2012; 7, 62). Although the results of their study are important, we still think that our study is informative, because we studied the effects of Olaparib on genomic instability and on the increase in cell migration and resistance associated with genome instability. In addition, we also found that Olaparib suppressed all those genomic instability-associated effects. 

Regarding cGAS/STING pathway activation, the reviewer remarked that “the evidence was only limited to the cytosolic cGAS observed in immunofluorescence microscopy”. However, this criticism is not merited, because we also provided supporting data showing the expression of downstream factors Ifnβ, Ifit1, and Ifit3.

The reviewer’s remark on repair pathways is related to major comment 2. Please see our response there.

We carefully rewrote the manuscript to address the issues raised by the reviewer, which we think has improved the manuscript. 

Major comments:

1. Olaparib was previously shown to disrupt cGAS STING pathway and in BRCA1 deficient cells (Miura et al Radiat Oncol, 2012; Ding et al Cell Rep, 2018). The Miura et al, Radiat Oncol paper should be referenced in this manuscript.

Response: Thank you for this suggestion. We have added references to those manuscripts and related background information.

2. PARP inhibitors are a classical example of a synthetic lethality relationship of diverging DNA repair mechanisms, and it is possible that Olaparib kills cancer cells after other HR pathways were compromised (eg. cells with stalled replication forks due to defective BRCA1 pathway). It was unclear which homologous repair pathway (eg BRCA1/Fanconi Anemia pathway) or non homologous end joining pathway (53BP1) that Olaparib affects. Cell imaging following Olaparib treatment with gH2AX and Rad51 staining should be repeated in 53BP1 and/or BRCA1/FANCD2 knock out cell lines. Alternatively, the authors could repeat the immunofluorescence imaging with BRCA1/FANCD2 to determine which pathway Olaparib affects.

Response: As pointed out, a PARP inhibitor is a drug that causes a synthetic lethal effect in HR deficient cells. However, in our manuscript, we did not use a PARP inhibitor for this purpose. We studied the effects of Olaparib on HR proficient cancer cells. 

In our experiments, the major type of DNA damage caused by CPT and ionizing radiation is replication stress-associated DSBs (Matsuno et al., iScience 2021; Atsumi et al., JBC 2012). Although DSBs can also directly result from irradiation, those are usually repaired within a few hours. Persistent DSBs that increase the risk of genomic instability occur during the S phase that follows irradiation and are associated with replication stress (Matsuno et al., iScience 2021), similar to the cells treated with CPT. The resulting genomic instability further increases the risk of cell migration (Matsuno et al., iScience 2021) and the clonal evolution of cells with increased resistance (Matsuno et al., Nature Com 2019); therefore, it is important to suppress the effects of genomic instability. 

Replication stress-associated DSBs are the prime target of homologous recombination (HR). Genomic instability is usually caused by erroneous DNA repair. Therefore, we simply tested the effects of PARP inhibitor, because PARP inhibitor effectively blocks the induction of HR (Hochegger et al., EMBO J. 2006; Sugimura et al., JCB 2008). The results were basically as expected, in which the killing effect was elevated. 

In our experiments, we monitored Rad51/gammaH2AX foci to see if HR-associated damage and 53BP1/gammaH2AX foci determine the resulting DSBs. We did not examine the involvement of BRCA1/FANCD2, because our study was not designed to address mechanism questions relating to the Fanconi pathway. In fact, our basic aim was to study cellular responses to replication stress-associated DSBs, not to ICL type DNA damage, which involves the Fanconi pathway.

We carefully rewrote the manuscript to address these issues, which we believe has made the manuscript clearer. 

3.Fig 5a – higher resolution of the figure containing the graph is needed. The number P=.. were ineligible. Fig 5c was missing two points for the gamma-ray irradiation (15 and 20 Gy).

Response: Thank you for this comment. We have improved the resolution of the images. The missing two points were because there were no surviving cells under this condition. The issue has been addressed in the legend.

4. Page 17 line 305 – there is already a paper that describes Olaparib association with BRCA or Fanconi Anemia pathway, please cite Ding et al Cell Rep. 2018 Dec 11; 25(11): 2972–2980.e5 as Olaparib has been shown to trigger cGas in BRCA1-deficient cells, which suggests it efficacy in killing cancer cells.

Response: We have added a reference to this paper. 

5. Please add the concentration of Olaparib as it was not described anywhere in the manuscript or methods.

Response: This information has been added to the Methods section.

Minor comments:1. page 18 line 308 is missing a reference after "..presumably by non-homologous end joining".

2. page 18 line 313 please add a sentence or two on how this work will bring impact to the cancer therapeutics or DNA damage fields.

3. Figure 4b remove the "With or without Olaparib" figure legend as the data was not shown in the figure. Alternatively, the authors could clarify that the graph on Fig 4b was without Olaparib.

4. Figures 2a and 3c was unclear. Higher resolution images are required to see the foci.

Responses to minor comments 1 to 4: We have addressed these issues in the revised manuscript.

 

Reviewer #2: Suzuki et al. presented a concise study on the suppression of radio- and CPT surviving cells by Olaparib co-treatment. Although the enhanced anti-tumour activity of Olaparib and CPT or gamma-rays was presented several years ago (Miura et al 2012 doi: 10.1186/1748-717X-7-62), the authors are addressing important aspect of simultaneous treatment that enhance elimination of resistant cancer cells, using MCF7 and to some extent HeLa and SW480 models. Overall, the text is well-written and data presented on figures are complete. Thus, I recommend the manuscript for publication, however, for increasing accessibility of the manuscript content I suggest some minor modifications of text, listed below.

Response: We very much appreciated for the reviewer’s encouraging comments. We also appreciated the other comments, which we have addressed on a one-to-one basis below. 

1) The quality of figures (at least in my version, even after download) was low and should be enhanced in final version – the authors should clearly export the files at least in 300 DPI.

Response: We have improved the quality of the figures. 

2) Fig 4 b and c – the authors claim that ‘no cells irradiated three times were obtained in the presence of Olaparib, because growth failed to recover in the presence of Olaparib’. This description is unclear to me and should be extended / commented. It is a bit disturbing that absolutely no cells were found in the presence of Olaparib. I convince the authors to rewrite the description of this results to explain the observed strong effects.

Response: We have carefully revised the text to address this comment.

3) Lines 272-273 require edition as it reads strange in current version.

Response: We have revised the text.

4) Lines 302-304 The authors should state here that it is simultaneous treatment of CPT or gamma-rays with Olaparib.

Response: We now include this statement.

5) The final sentence of Discussion (lines 311-313) should be followed by a major conclusion on the findings described in this work to leave the reader with a clear message.

Response: We have added conclusion statements to the final paragraph. 

6) Methods – line 319 – used or prepared?

Response: We have corrected this.

---

## [Decision Letter · Decision Letter 1]

17 Jan 2023

Genome Destabilization-Associated Phenotypes Arising as a Consequence of Therapeutic Treatment are Suppressed by Olaparib

PONE-D-22-25618R1

Dear Dr. Yoshioka,

We’re pleased to inform you that your manuscript has been judged scientifically suitable for publication and will be formally accepted for publication once it meets all outstanding technical requirements.

Kind regards,

Alvaro Galli

Academic Editor

PLOS ONE

Additional Editor Comments (optional):

Reviewers' comments:

Reviewer's Responses to Questions

**Comments to the Author**

1. If the authors have adequately addressed your comments raised in a previous round of review and you feel that this manuscript is now acceptable for publication, you may indicate that here to bypass the “Comments to the Author” section, enter your conflict of interest statement in the “Confidential to Editor” section, and submit your "Accept" recommendation.

Reviewer #1: All comments have been addressed

Reviewer #2: All comments have been addressed

2. Is the manuscript technically sound, and do the data support the conclusions?

Reviewer #1: Yes

Reviewer #2: Yes

3. Has the statistical analysis been performed appropriately and rigorously? 

Reviewer #1: Yes

Reviewer #2: Yes

4. Have the authors made all data underlying the findings in their manuscript fully available?

Reviewer #1: Yes

Reviewer #2: Yes

5. Is the manuscript presented in an intelligible fashion and written in standard English?

Reviewer #1: Yes

Reviewer #2: Yes

6. Review Comments to the Author

Reviewer #1: (No Response)

Reviewer #2: In current version, the authors have addressed all the points raised in the first round of review.

Present text reads well, Line 151 requires correction: stability should be replaced by instability. Overall, the manuscript is now ready for publication.

7. PLOS authors have the option to publish the peer review history of their article (what does this mean?). If published, this will include your full peer review and any attached files.

Reviewer #1: **Yes: **Winnie Tan

Reviewer #2: No

---

## [Editor Report · Acceptance letter]

19 Jan 2023

PONE-D-22-25618R1 

Genome Destabilization-Associated Phenotypes Arising as a Consequence of Therapeutic Treatment are Suppressed by Olaparib 

Dear Dr. Yoshioka:

I'm pleased to inform you that your manuscript has been deemed suitable for publication in PLOS ONE. Congratulations! Your manuscript is now with our production department. 

Kind regards, 

on behalf of

Dr. Alvaro Galli 

Academic Editor

PLOS ONE